# Angiogenetic Factors in Chronic Subdural Hematoma Development

**DOI:** 10.3390/diagnostics12112787

**Published:** 2022-11-14

**Authors:** Andrey Petrov, Arkady Ivanov, Natalia Dryagina, Anna Petrova, Konstantin Samochernykh, Larisa Rozhchenko

**Affiliations:** Vascular Neurosurgery Department, Polenov Neurosurgical Research Institute, Branch of Almazov National Medical Research Centre, 191014 St. Petersburg, Russia

**Keywords:** chronic subdural hematoma (CSDH), vascular endothelial growth factor (VEGF), matrix metallopeptidase 9 (MMP9), angiopoietin-2 (Ang2), transforming growth factor beta 1 (TGF-β1), platelet-derived growth factor BB (PDGF-BB), endovascular embolization, squid, middle meningeal artery (MMA), ethylene-vinyl alcohol copolymer

## Abstract

The levels of angiogenic factors were analyzed in eight patients who underwent the embolization of chronic subdural hematoma (CSDH) with non-adhesive liquid embolic agents. Four of these patients had previously undergone surgical treatment for hematoma removal and had recurrences of a similar volume, and four had an increase in hematoma volume due to rebleeding. The levels of vascular endothelial growth factor (VEGF), matrix metallopeptidase 9 (MMP 9), angiopoietin-2 (Ang2), transforming growth factor beta 1 (TGF-β1) and platelet-derived growth factor BB (PDGF-BB) in the arterial and venous blood were analyzed. The most significant results were obtained from the peripheral venous blood samples. The levels of VEGF in the samples of all the patients were close to normal or slightly decreased. There was an increase in the MMP9 levels (the factor that contributes to the disintegration of the vessel wall components) in all the patients. The Ang2 and especially the PDGF TGF-β1 (the factor that plays an important role in the growth of the vessel wall from the already existing blood vessel tissue) levels were distinctly low in most of the cases and slightly elevated only in a number of patients who had previously been operated on. The results obtained show that there is an imbalance in the angiogenesis factors in patients with rebleeding CSDH. At the same time, the factors determining the formation of the vessel wall were reduced, and the levels of factors contributing to the degradation of extracellular matrix components were significantly increased. Such factors could help us to anticipate the increased risk of hemorrhages. Highlights: The levels of VEGF, MMP 9, Ang2, TGF-β1 and PDGF-BB in the arterial and venous blood were analyzed. The most significant results were obtained from the peripheral venous blood samples. The results obtained show that there is an imbalance in the angiogenesis factors in patients with rebleeding CSDH. Such a profile of factors could help us to anticipate the increased risk of hemorrhages.

## 1. Introduction

The term “chronic” subdural hematoma (CSDH) is used to refer to hematomas whose clinical manifestations are noted later than 3 weeks after the alleged event that caused them [1]. External factors, such as mild blunt trauma, anticoagulation/antiplatelet agents and excess alcohol consumption, and internal factors, such as arterial hypertension, cerebrovascular atherosclerosis, diabetes mellitus and brain atrophy, are generally considered as predisposing factors for CSDH.

CSDH is estimated to occur in 2–20 per 100,000 people in the general population each year and increases with age, with a high frequency of about 50–60 per 100,000 people aged 70 years and above. CSDH is considered as one of the most common diseases in neurosurgery [2].

Generally, the etiology of CSDH has been multifactorial, and it has never been caused by trauma alone [3]. R. Virchow, in 1857, hypothesized that organized exudate accumulates in the subdural space due to a generalized inflammatory process based on his pathoanatomical studies [4,5,6]. Successive studies have demonstrated that the accumulation of blood in the subdural space induces a nonspecific inflammatory reaction after a traumatic brain injury and, ultimately, leads to the formation of a vascular neo-membrane, which is responsible for recurrent micro bleeds [5,6,7].

Membrane formation is an important aspect of the pathophysiology of CSDH. A possible explanation for the increase in CSDH due to repeated hemorrhages is microvascular bleeding within the membrane. The growth of the vasculature begins in the distal branches of the regional artery (i.e., in the vast majority of cases, it is a middle meningeal artery (MMA)) [8,9,10,11,12,13,14,15].

Vascular growth is determined by the balance between its stimulants and inhibitors. The process of vessel maturation involves a stepwise transition from an actively growing vascular bed to a resting, fully formed and functional network. In this case, the suppression of endothelial proliferation, the appearance of new capillaries, the stabilization of existing vascular tubules and the incorporation of cells that form the vessel walls occur [16].

The most important regulator of both physiological and pathological angiogenesis is the vascular endothelial growth factor (VEGF), which stimulates the growth and division of endothelial cells in the formation of a primary capillary [17].

Next comes the stage of the maturation of the vessel—the attraction of the cells that form the walls of blood vessels—along with the pericytes and smooth muscle cells (SMCs) [18]. Several different factors are involved in the recruitment of pericytes to form the walls of newly formed vessels, but platelet-derived growth factor (PDGF) plays a key role in this process. PDGF is an important mitogen for various mesenchymal cell types, such as fibroblasts, SMCs and pericytes. During angiogenesis, the increased expression of PDGF-β messenger ribonucleic acid in the tip cells causes their proliferation, directed migration and incorporation into the vessel wall of the pericytes, on the surface of which the platelet-derived growth factor receptor β (PDGFR-β) is expressed. At the same time, for the proper formation of the vessel wall, it is important that PDGF-β is expressed precisely by the endothelial cells [16,18,19].

Transforming growth factor β (TGF-β) is activated when contact occurs between endothelial cells and mesenchymal progenitor cells of pericytes [20].

Another signaling system is involved in the regulation of the complex interactions between the endothelium and surrounding cells: the tyrosine kinase receptor Tie2, expressed by the endothelial cells and their ligands angiopoietins (Ang 1 and 2) [21]. Ang2 suppression of the Tie2 receptor signaling pathway inhibits pericyte influx, impairs vessel maturation and renders vessels hyper-responsive to the stimulatory effects of VEGF.

A key consequence of VEGF and Ang2 signaling Is the expression of matrix metalloproteinase-9 (MMP9) [19]. Matrix metalloproteinases-9 (MMP9) has a significant degrading capacity for almost all the components of the extracellular matrix, increasing the risk of bleeding.

MMA embolization is a method enabling the devascularization of the CSDH membrane. MMA embolization results in a lower recurrence rate of 3.6% [8,10]. It is believed that MMA embolization interferes with the final stage of the pathophysiological cascade (cycle) of the formation of CSDH [22,23,24].

The standard practice for patients with clinical manifestations of hematoma is the evacuation of CSDH. Burr hole and twist drill craniostomy and craniotomy yield similar results in terms of their safety and efficacy [25,26]. In 10–30% of operated patients, a recurrence of CSDH occurs [27,28]. Additionally, with bilateral CSDH, the recurrence rate is even higher than that of unilateral CSDH [9,27,29,30,31,32,33,34,35,36,37,38,39].

Despite this great interest, there are many unresolved issues affecting the pathogenesis and, accordingly, the tactics for treating this disease. The open questions in this context are:-What are the concentrations of the leading factors in neo-angiogenesis directly in the arterial blood in the vessels that feed the outer membrane of the CSDH capsule (MMA branches), and do they differ from the concentrations in the peripheral veins?-Are there any differences in the concentrations of angiogenesis factors in the bloods between patients who have repeated hemorrhages in the area of CSDH (rebleeding) and patients with a fairly homogeneous structure of the hematoma according to non-contrast computed tomography (NCCT)?-Are the hematoma volume, recidivation and side location related to the concentrations of angiogenesis factors in the blood?

## 2. Materials and Methods

### 2.1. Study Design

This is a case–control, non-randomized, monocentric study. The case series was designed to study the angiogenesis factors in chronic subdural hematoma development. This article presents the second case series (the first series was published in 2021 [3,40]) based on the sole use of Squid (Balt Extrusion) for middle meningeal arterial (MMA) embolization as a treatment option for CSHD. The choice of Squid rather than Onyx (Medtronic) was due to the availability of this low-viscosity option [41], which allows for a deeper penetration of the embolizing agent into the vessels feeding the CSDH capsule [3,40]. In all the patients, we determined the concentrations of angiogenesis factors in the peripheral venous blood and in the MMA (arterial blood).

The study was conducted at the Almazov National Research Medical Center. Patients were enrolled in the study from November 2021 to May 2022.

### 2.2. Adherence to Ethical Standards

Approval for the study was granted and monitored by the local ethics committee of the National Research Medical Center, Almazov. The research was conducted in accordance with the 2013 version of Helsinki Declaration by the World Medical Association. All patients involved in the study were informed about the nature of the study, the treatment and the odds and potential risks associated with it.

### 2.3. Participants and Enrolment Criteria

Our case series studying the angiogenesis factors included 8 patients over 18 years of age with the presence of CSDH, and the average age of the patients was 61 ± 14 (m ± SD). All the selected patients underwent endovascular embolization of the MMA branches. The study did not include pregnant patients or patients who had blood diseases, bronchial asthma, diabetes mellitus, oncological diseases or rheumatological diseases, as well as patients taking anticoagulants and antiplatelet agents.

For the control group (*n* = 33), we conducted a study of the levels of the angiogenesis factors in the peripheral venous blood in conditionally healthy volunteers. This group included people over 18 years old who signed an agreement to participate in the study.

### 2.4. Variables

The differences in the levels of angiogenesis factors in the arterial blood obtained by MMA embolization with the levels in the peripheral venous blood from the same patients were determined. The associations of the angiogenesis factors in the peripheral venous blood and arterial blood with the side location of CSDH, rebleeding status, surgery status before MMA embolization and total pre-embolization CSDH volume were also studied.

For the collection of anamnesis, a retrospective study of the disease history was conducted, and in the assessment of the NCCT head scans, special attention was paid to the surgical status before MMA embolization, the CSDH side location and rebleeding status.

#### 2.4.1. Volume Measurement of Chronic Subdural Hematoma

The volume of the CSDH was determined by manually measuring the specific volumes, defining the area of interest using OsiriX software (Osirix for Mac, version 11.0) on a pre-embolization of NCCT [3]. In cases where the hematoma was located bilaterally, the hematoma volumes were summed and a total hematoma volume was obtained.

Bilateral CSDH (2/8 patients) required the bilateral embolization of the MMA and, therefore, 10 embolization procedures were performed on 8 patients. (Table 1).

#### 2.4.2. Determination of the Concentrations of Angiogenesis Factors

All eight patients underwent endovascular embolization of the corresponding MMA. All operations were performed under general anesthesia and were grouped into two sets. First, in the group of patients with unilateral hematoma, 6 patients underwent embolization of the frontal branch of the middle meningeal artery and capsule vessels from the hematoma side, while in the second group with bilateral CSHD, 2 patients underwent the sequential embolization of the MMA on both sides. Blood for the research was taken from a peripheral vein (*n* = 8) before surgery (15 min before general anesthesia) and during surgery from the main trunk of the MMA (*n* = 7) using a 21” microcatheter (Figure 1).

Clotting activator vacutainers were used to obtain serum, and EDTA vacutainers were used to obtain plasma. The clotting activator tubes were left at room temperature to form a clot for 1 h and then centrifuged for 30 min at 3000 rpm. The EDTA tubes were centrifuged immediately upon arrival at the laboratory for 30 min at 3000 rpm at room temperature. After centrifugation, the serum and plasma were collected, aliquoted, frozen and stored at −20 °C until the time of examination. The concentrations of the angiogenesis factors were determined by enzyme immunoassay using a Personal Lab automatic plate analyzer (Italy). The concentrations of vascular endothelial growth factor (VEGF), matrix metalloproteinase-9 (MMP-9), angiopoietin-2 (Ang2) and platelet-derived growth factor-β (PDGF-β) were determined using kits from R&D Systems (USA), and the concentration of transforming growth factor-β1 (TGF-β1) was determined using a kit from Invitrogen (Austria). The study of VEGF, MMP-9 and Ang2 was performed using blood serum obtained from a vein, including PDGF-β and TGF-β1. The reference values, according to reagent kit manufacturers, were 62–707 pg/mL for VEGF; 169–705 ng/mL for MMP-9; 1065–8907 pg/mL for angio-2; 942–7366 pg/mL for PDGF-β; and 5222–13,731 pg/mL for TGF-β1. We conducted a study of the levels of angiogenesis factors in the venous blood of 33 healthy volunteers, and their average age was 54 ± 15 (m ± SD) (male—20, female—13). We determined the average levels of VEGF as (*n* = 33)—271 ± 41 (m ± SD) pg/mL, MMP-9 as (*n* = 33)—453 (Me)ng/mL and Ang 2 as (*n* = 33)—2198 ± 248 (m ± SD) pg/mL, as well as TGF-β1 as (*n* = 21)—10,936 ± 411 (m ± SD) pg/mL and PDGF-β as (*n* = 21)—3611 ± 371 (m ± SD) pg/mL. These data coincided with the range of the reference values according to the manufacturer based on the test systems of the healthy volunteers. These values were taken as the control values. We detected whether the levels of angiogenesis factors in the blood were exceeded by determining the output of the limits of the standard deviation from the results obtained from the group of healthy volunteers. Data on the angiogenesis factors obtained from the arterial blood were compared with the data on the concentrations of angiogenesis factors obtained from the venous blood.

### 2.5. Statistical Methods

A statistical analysis was performed using StatTech v. 2.8.8 (Developer—StatTech LLC, Moscow, Russia, the software is registered by the Federal Service for Intellectual Property, accession number 2020615715, registration date 29 May 2020).

Quantitative variables were assessed for normality using the Shapiro–Wilk test (when the number of subjects was less than 50) or the Kolmogorov–Smirnov test (when the number of subjects was more than 50).

Quantitative variables following a normal distribution were described using the mean (M) and standard deviation (SD), and the 95% confidence interval (95% CI) for the mean was estimated.

Quantitative variables following a non-normal distribution were described using the median (Me) and lower and upper quartiles (Q1–Q3).

The comparison of the two groups, for a quantitative variable following a normal distribution, was performed using Student’s *t*-test if the variances were equal and Welch’s *t*-test in the case of unequal variances.

The Mann–Whitney U-test was used to compare the two groups in the case of a quantitative variable whose distribution differed from the normal distribution.

The direction and strength of the association between two quantitative variables were estimated using Pearson’s correlation coefficient (in the case of a normal distribution of the variables). The direction and strength of the association between two quantitative indicators were estimated using Spearman’s correlation coefficient.

The prognostic model characterizing the dependence of a quantitative variable on the predictors was developed using ordinary least squares linear regression.

ROC analysis was used to assess the diagnostic performance of the quantitative variables in predicting a categorical outcome. The optimal cut-off value of the quantitative variable was estimated using Youden’s J statistic. A full statistical analysis can be seen in the Appendix A.

## 3. Results

Men predominated in the study (7:1), accounting for five out of the eight patients who had previously suffered a traumatic brain injury. Our patients did not receive antithrombotic or anticoagulant therapy.

The characteristic of our study was that out of all our patients, four had previously been operated on, but without any radiological improvements. Three were potential candidates for open surgery aiming to remove a hematoma in a case of progressive neurological deterioration. Four CT scans of four patients showed signs of repeated hemorrhages before embolization (Figure 2).

When analyzing the angiogenesis factors obtained from the venous and arterial bloods from the patients with CSDH, the following results were found (Table 2):

In accordance with the findings presented in Table 2, when comparing the VEGF (vein) Angio-2 (vein), TGF-β1 (vein) and PDGF-β (vein), statistically significant differences were revealed in comparison with the healthy volunteers. However, MMP-9 (vein) did not show statistically significant differences in comparison with healthy volunteers (*p* = 0.278).

Respectively, the level of VEGF in seven out of the eight patients was significantly (*p* = 0.004) reduced.

In six out of the eight patients, Ang2 was higher than the level of the healthy volunteers, and it turned out to be lower in two non-operated patients.

Transforming growth factor TGF-β1 was significantly (*p* = 0.026) reduced in all the examined patients. The mean level of TGF-β1 in the venous blood of patients with CSDH was 7462 ± 3497 pg/mL. It was significantly lower compared to the level obtained from the venous blood of healthy volunteers, being 10,936 ± 411 pg/mL. TGF-β1 in all the patients was below the level in the healthy volunteers.

The mean level of PDGF-β in the venous blood of the patients with CSDH was 2372 ± 1371 pg/mL, which was significantly (*p* = 0.038) lower compared to the level obtained from the venous blood of healthy volunteers, 3611 ± 371 pg/mL.

The mean concentrations (M ± SD/Me) in the arterial blood samples (*n* = 7) were 37 (pg/mL) for VEGF, 596 ± 416 (ng/mL) for MMP-9, 3799 ± 2259 (pg/mL) for Angio-2, 8869 ± 1327 (pg/mL) for TGF-β1 and 2683 ± 1042 (pg/mL) for PDGFβ-.

The strength of the association was assessed using the Chaddock scale. A close (VEGF (vein/artery); TGF-β1 (vein/artery); PDGF-β (vein/artery)), strong (MMP-9 (vein/artery)) and functional (Angio-2 (vein/artery)) positive association between the levels of angiogenesis factors in the patient’s venous blood and the levels in the patient’s arterial blood was identified (Figure 3).

We did not find any statistically significant differences in the concentrations of VEGF, MMP-9, Angio-2, TGF-β1 and PDGF-β in the venous and arterial blood samples of patients depending on the side of the CSDH.

Only the concentration levels of MMP-9 were statistically significantly elevated in both the venous (*p* = 0.021) and arterial (*p* = 0.025) blood of patients who were diagnosed rebleeding based on the NCCT head scans (Figure 4 and Figure 5).

The levels of MMP9 (venous and arterial blood samples) were significantly elevated (by 2.5 times) in four rebleeding CSDH patients out of the total eight.

An interesting relationship was found when assessing the association between the concentration levels of angiogenesis factors and the total hematoma volume. According to the Chaddock scale, a strong (VEGF (*p* = 0.040, *p* = 0.033)) positive correlation was found between the concentrations in the venous and arterial blood samples of the patients and the total pre-embolization volume of CSDH. In a similar situation, in the case of MMP-9, a close (*p* = 0.056,), strong positive (*p* = 0.068) correlation with the total pre-embolization volume CSDHs was found. A moderate, close positive correlation between TGF-β1 in the venous and arterial blood samples and total pre-embolization volume of the CSDHs was found (*p* = 0.257, *p* = 0.248). A close, weak positive correlation between PDGF-β in the venous and arterial blood samples of the patients and total pre-embolization volume of the CSDHs was estimated. (*p* = 0.204, *p* = 0.585). On the contrary, there was no association between the total pre-embolization volume of the CSDHs and Angio-2 (vein, artery) (*p* = 0.778) (*p* = 0.939).

When comparing the concentrations of VEGF, MMP-9, Angio-2, TGF-β1 and PDGF-β in the venous and arterial blood samples of the patients depending on the surgery status, no statistically significant differences were revealed.

## 4. Discussion

Thus, the level of VEGF in seven out of eight patients was significantly reduced and averaged 271 ± 41 pg/mL compared with the level obtained from the venous blood of the healthy volunteers, which may indicate the completion of the growth of the endothelium of the newly formed vessels and the most pronounced changes in the levels of the factors responsible for the proper maturation of the vascular wall.

Normally, in the mature vasculature, the production of Ang2 is suppressed, and its overexpression, as observed in the all patients in our series, ensures the formation of abnormally dilated vessels, without a mature wall structure. Ang2, which accelerates the spread and migration of endothelial cells, thus stimulates the germination of new blood vessels [18,42].

The regulation of angiogenesis largely depends on the balance between the factors that stimulate or suppress certain levels of the vascular network formation. Thus, Ang2 acts as an angiogenesis-initiating signal that has an activating effect on the other factors involved in vessel formation. A key consequence of VEGF and Ang2 activation signaling is the matrix metalloproteinase (MMP) expression. Matrix metalloproteinases-9 (MMP9) has a significant degrading ability with respect to almost all the components of the extracellular matrix, with an active participation in the remodeling of the vasculature at the initial stage [19,43].

In four out of the eight patients, the MMP9 level was sharply elevated, and this correlated with the group of patients with rebleeding. The violation of the correct formation of the walls causes an increase in vascular permeability and expansion (dilation), which subsequently leads to edema and hemorrhage. It is the maturation stage associated with the formation of the walls of newly formed vessels that is disturbed in CSDH, which consists of poorly organized, immature, super-permeable vessels prone to hemorrhages [19,31,44].

The activation of TGF-β leads to the inhibition of the proliferation and migration of the endothelial cells, the suppression of the expression of their VEGFR2 receptor and the differentiation of progenitor cells into pericytes [20]. This leads to the formation of a full-fledged normal vessel, while there is a decrease in the level of TGF-β, as observed in all the patients in our series. Transforming growth factor TGF-β1 was significantly reduced in all the examined patients. A low level of transforming growth factor indicates the absence of TGF-β1 induction and the differentiation of progenitor cells into pericytes and is one of the reasons for the formation of an immature leaky network of super-permeable neo-membrane vessels.

Microvascular integrity can be compromised when the PDGF-β expression is too high or too low. A decrease in the level of PDGF-β leads to a deficiency in SMCs and pericytes, the destruction of the blood–brain barrier (BBB), increased permeability of the circulating plasma proteins moving to the brain and bleeding. The number of pericytes is inversely correlated with the degree of the overt symptomatic hemorrhage or clinically occult micro-hemorrhage [16,18].

In our series, the majority of patients showed a decrease in the level of PDGF-β. According to our data, the level of PDGF-BB was reduced in six out of the eight patients with CSDH, which may explain the development of bleeding. The number of pericytes was inversely correlated with the degree of the overt symptomatic hemorrhage or clinically occult micro-hemorrhage.

As the subdural hematoma becomes chronic, numerous cycles of fibrin formation and fibrinolysis occur, and an internal neo-membrane with high-density septa may develop [45,46,47]. The histological examination of the CSDH multilayer membrane revealed giant capillaries and macrophage infiltration in the outer and inner layers; newly formed capillaries with highly permeable endothelial gap junctions, expressing high levels of vascular endothelial growth factor (VEGF) and placental growth factor (PlGF); proliferating fibroblasts, forming fibrous granulation tissue with collagen deposition; chronic lymphoplasmacytic and histiocytic inflammation; and macrophages containing hemosiderin [46,48]. Accordingly, studies that allow us not only to clarify the role of certain angiogenesis factors involved in the pathogenesis of CSDH but also to determine the methods for their correction are of particular relevance [49,50,51].

A close, strong and moderate positive correlation was found in both the venous and arterial blood samples between the total pre-embolization volume of the CSDHs and the concentration levels of VEGF, MMP-9, TGF-β and PDGF-β. Thus, the larger the total volume of the hematoma was, the higher the concentrations of these angiogenesis factors were. In 2014, R. Weigel [52] described a significant correlation between the concentrations of VEGF in hematoma fluid of different classes based on the CT appearance and the respective exudation rates previously reported by M. Tokmak [53]. F. Li et al. (2017) [54] found a correlation between the volume of “fresh” blood in CSDH on brain MRI and the VEGF concentration levels in both the venous blood and hematoma contents. In our series, the level of VEGF concentration in the venous blood was significantly lower than that in the healthy volunteers. However, there was also a tendency towards an increase in the number of angiogenesis factors in the blood depending on the volume of the hematoma. Perhaps the marker of repeated bleeding is MMP-9, and the volume of “fresh” blood directly correlates with the other factors, particularly VEGF. In the study of the angiogenesis factors in the arterial blood, the same trends as those in the venous blood were observed. However, this requires further study due to the small number of observations and the lack of reference values.

## 5. Conclusions

The excess levels of angiogenesis factors in the venous blood of patients as opposed to healthy volunteers, as revealed by us, are proportional to the volume of the hematoma and correlate with the presence of repeated hemorrhages. Thus, our data indicate a disruption of the process of the formation of healthy vessels with a full wall in the neo-membrane in patients with CSDH due to a significant imbalance in the angiogenesis factors responsible for this process. The identification of such disorders can not only explain the molecular basis for the formation of the vascularization of the CSDH capsule but also affect treatment strategies in the future.

## Figures and Tables

**Figure 1 diagnostics-12-02787-f001:**
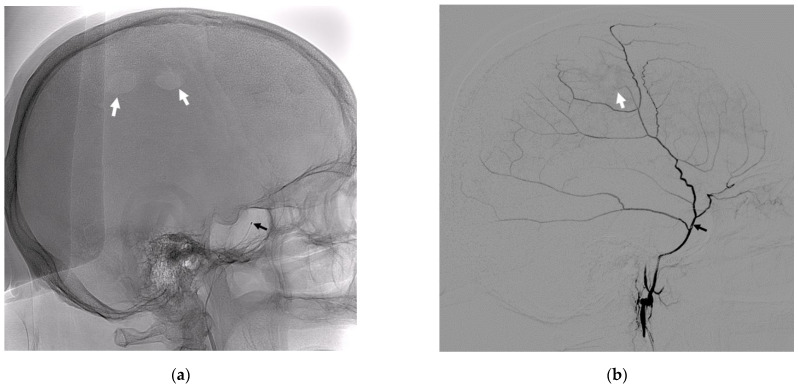
Lateral radiographic images without subtraction (**a**) and with subtraction (**b**). (**a**) The position of the 21” microcatheter in the MMA trunk (black arrow), and the white arrows indicate the radiological traces of the burr hole trepanation. (**b**) A lateral subtraction angiogram of the main trunk and branches of the MMA, where the black arrow indicates the position of the microcatheter, and the white arrow indicates the disturbed zone of the vascularization of the dura mater in the region of the burr hole trepanation.

**Figure 2 diagnostics-12-02787-f002:**
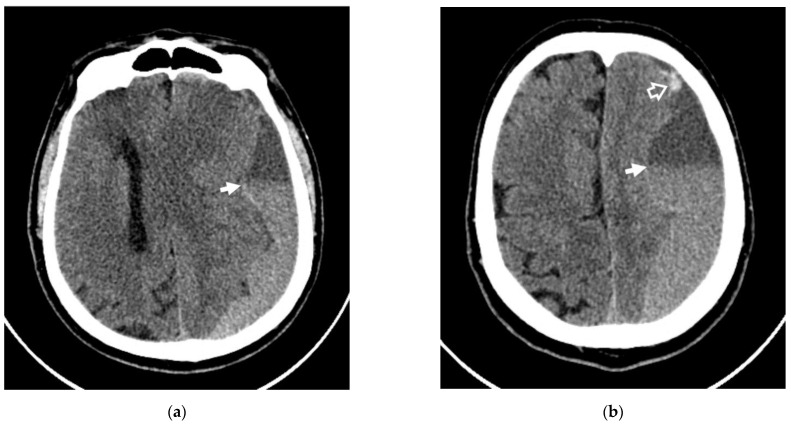
Axial NCCT scans of patient #1, showing rebleeding in the area of CSDH. (**a**,**b**) White arrows indicate the level of sedimentation of the fresh blood in the area of CSDH; (**b**) white frame arrow denotes the rebleeding area.

**Figure 3 diagnostics-12-02787-f003:**
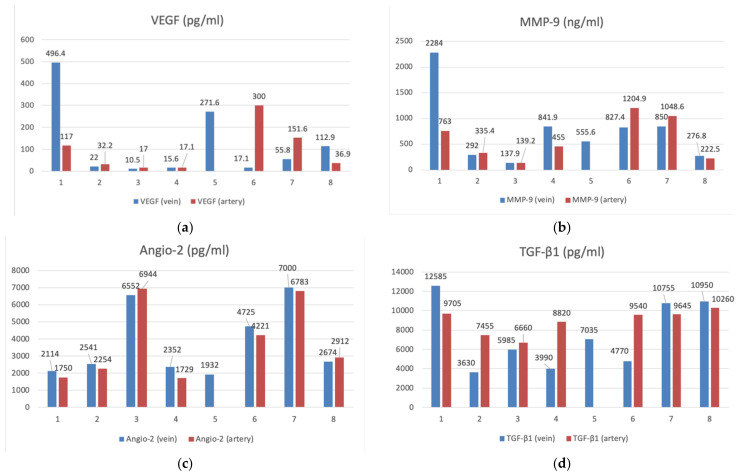
The bar graphs with datapoints show close trends in the concentrations of angiogenesis factors in the peripheral venous (*n* = 8) and arterial (*n* = 7) blood. (**a**) VEGF (pg/mL), (**b**) MMP-9 (ng/mL), (**c**) Angio-2 (pg/mL), (**d**) TGF-β1 (pg/mL), (**e**) PDGFβ- (pg/mL).

**Figure 4 diagnostics-12-02787-f004:**
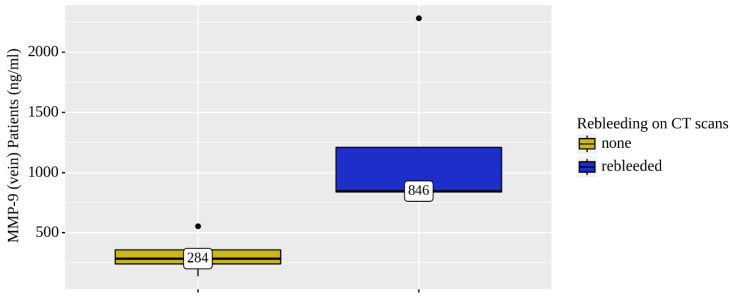
Analysis of MMP-9 (vein) patient (*n* = 8) rebleeding conditions on NCCT brain scans.

**Figure 5 diagnostics-12-02787-f005:**
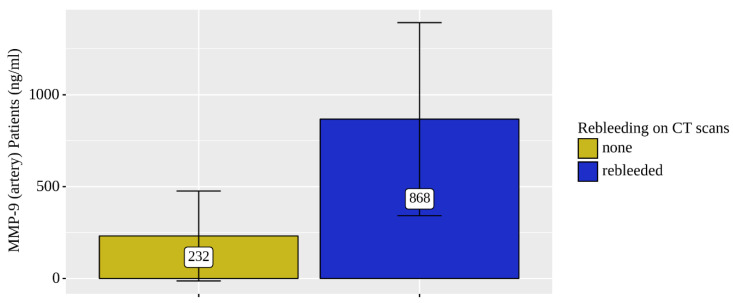
Analysis of MMP-9 (artery) patient (*n* = 7) rebleeding conditions on NCCT brain scans.

**Table 1 diagnostics-12-02787-t001:** Summary of the demographic and treatment data of 8 patients with CSDH treated with Squid embolization of the MMA.

Patient # Age (Years) Gender	Side	Rebleeding on NCCT Scans	Surgery Status before MMA Embolization	Volume Pre-Embol (mL)	Total Volume Pre-Embol (mL)
**1, 57, m**	Left unilateral	rebled	none	170	170
**2, 66, m**	Left unilateral	none	Pre-embol left-side craniectomy	70	70
**3, 43, m**	Right unilateral	none	none	53	53
**4, 60, m**	Left unilateral	rebled	none	57	57
**5, 44, f**	Bilateral	none	none	right 44 left 54	98
**6, 58, m**	Left unilateral	rebled	pre-embol burr hole	70	70
**7, 82, m**	Bilateral	rebled	pre-embol right- and left-side burr hole	right 170 left 67	237
**8, 77, m**	Right unilateral	none	pre-embol burr hole	65	65

**Table 2 diagnostics-12-02787-t002:** Analysis of angiogenetic factors in patients/healthy volunteers and results of the correlation analysis of the association between the patients’ venous/artery blood samples.

	Mean (M ± SD/Me) Healthy Volunteers (*n*)	Mean (M ± SD/Me) Patients (*n*)	Comparison Patients’ Venous Blood Samples/Healthy Volunteers’ Venous Blood Samples (Applied Method)
	venous blood	venous blood	
VEGF (pg/mL)	271 ± 41 (*n* = 33)	39 (*n* = 8)	***p* = 0.004** * (Mann–Whitney U-test)
MMP-9 (ng/mL)	432 ± 48 (*n* = 33)	692 (*n* = 8)	*p* = 0.278 (Mann–Whitney U-test)
Angio-2 (pg/mL)	2198 ± 248 (*n* = 33)	2608 (*n* = 8)	***p* = 0.023** * (Mann–Whitney U-test)
TGF-β1 (pg/mL)	10,936 ± 411 (*n* = 21)	7462 ± 3497 (*n* = 8)	***p* = 0.026** * (Welch’s *t*-test)
PDGFβ- (pg/mL)	3611 ± 371 (*n* = 21)	2372 ± 1371 (*n* = 8)	***p* = 0.038** * (Welch’s *t*-test).

*—Differences are statistically significant (*p* < 0.05).

## Data Availability

The data presented in this study are available on request from the first author. The data are not publicly available due to patient privacy protection.

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
