# Peer review of "Angiogenetic Factors in Chronic Subdural Hematoma Development"

_diagnostics, 2022, doi:10.3390/diagnostics12112787_

Round 1
Reviewer 1 Report
The authors present an interesting study concept contributing to a better understanding of the pathogenesis of chronic subdural hematoma (cSDH), a very common, and with a growing population of elderly patients, a highly relevant topic. However, there are fundamental methodological and structural issues in the current manuscript:
1. The introduction is way too long and very extensive, and the reader cannot really follow, what the authors want to study.
2. The result section does describe irrelevant clinical / surgical outcome, and not any word on the angiogenesis factors obtained from the venous blood samples of the cSDH patients investigated. This is confusing.
3. The results should at least be compared to a cohort of healthy volunteers with a similar cohort size, and the focus must be clear on laboratory investigations rather than clinical course and surgical / embolization outcome.
4. The tables provided are not very helpful and do not contain valuable information. The baseline table is way to detailed with clinical / radiological information, while the manuscript and discussion focuses on angiogenesis / pathophysiology. Table 2 provides numbers of factor serum levels without any reference, comparison or content that makes sense. Both of these tables are ok in a supplemental content, and more condensed and informational tables need to be in the manuscript.
5. Parts of the extensive content from the introduction are better fitted within the discussion. The authors need to shorten, sharpen and focus their work on angiogenesis factors and the laboratory investigations ONLY, without necessary clinical / radiological / interventional content.
6. The authors should also adhere to the published STROBE criteria for case series.
7. Why do the authors include a PubMed Search Result section in the introduction? This doesn't make a lot of sense in the context of the provided study design and common structure of scientific papers.
Please fundamentally re-strucutre and re-design this study. I am happy to read at the revised version.
Author Response
Thank you for revision. We Took into account all your comments in the lattest Version
Reviewer 2 Report
This is an interesting study, however the way the data is presented needs to be greatly improved.
- Shorten the abstract, add time of blood sampling and remove highlights
- Overall the introduction is much to long, this needs to be reduced by at least 75%. This is not a review, but a case report.
- Did patients give written consent
- When were blood samples taken? Was it the same for all patients?
- line 247-248 is truncated. Please complete sentence.
- ELISA data needs to be presented as a bar graph with datapoints
- Please discuss data in the results section and not in the methods
Author Response
Thank you for revision. All your comments and suggestions were taken into account in the lattest Version
Round 2
Reviewer 1 Report
The authors did not follow my recommendations of adequately adapting the manuscript following the STROBE criteria, focusing the research question and eliminating / adapting the tables with relevant content. Still, the introduction is way to extensive and the overall focus of this manuscript is not clear to me.
Author Response
The authors are very grateful to the referee for a detailed review of our article and useful recommendations.
We have tried to shorten the introduction and focus on angiogenesis. It can be further reduced if needed.
Conducted a more detailed statistical analysis of the data and redesigned the presentation of the results accordingly. The discussion focused on angiogenesis factors.
Part of the data and tables not included in the article is in the shot supplimentary file. Complete statistical analytics in a full supplementary file. Changes to become according to STROBE made it possible to structure the text. Once again, we express our gratitude and are ready to improve the article if necessary.

Reviewer 2 Report
The authors responded to my questions and modified their paper appropriately. Please add SD in the table for the mean of patients.
Author Response
The authors are grateful for the review and useful recommendations. According to your advice, we have tried to represent the ELISA data in the form of bar graphs. Added statistical analysis and focused on angiogenesis factors. As per your recommendations, we tried to state the results more clearly. Once again we express our gratitude and are ready for further correction.

Round 3
Reviewer 1 Report
The authors adressed my raised issues and focused on angiogenetic factors only. They moved less relevant information into the Supplemental Material. The manuscript, although still beeing too long overall, can be accepted in this version.